# What Do We Know about Thromboprophylaxis and Its Monitoring in Critically Ill Patients?

**DOI:** 10.3390/biomedicines9080864

**Published:** 2021-07-22

**Authors:** Philippe Cauchie, Michael Piagnerelli

**Affiliations:** 1Clinical Biology and Oncology-Hematology Department, CHU de Charleroi, Université Libre de Bruxelles, 6042 Charleroi, Belgium; 2Intensive Care, CHU de Charleroi, Université Libre de Bruxelles, 6042 Charleroi, Belgium; michael.piagnerelli@chu-charleroi.be; 3Laboratory of Experimental Medicine (ULB 222), Faculty of Medicine, Université Libre de Bruxelles, 6110 Montigny-le-Tilleul, Belgium

**Keywords:** venous thromboembolism, intensive care unit, anti-Xa activity, anticoagulation monitoring

## Abstract

Venous thromboembolism (VTE), including deep vein thrombosis and pulmonary embolism, is an important complication in patients hospitalized in intensive care units (ICU). Thromboprophylaxis is mainly performed with Low Molecular Weight Heparin (LMWH) and, in some specific patients, with Unfractionated Heparin (UFH). These intensive units are an environment where individual patient variability is extreme and where traditional antithrombotic protocols are frequently ineffective. This was known for a long time, but the hospitalization of many patients with COVID-19 inflammatory storms suddenly highlighted this knowledge. It is therefore reasonable to propose variable antithrombotic prevention protocols based initially on a series of individual criteria (weight, BMI, and thrombotic risks). Secondly, they should be adjusted by the monitoring of anticoagulant activity, preferably by measuring the anti-Xa activity. However, we still face unresolved questions, such as once- or twice-daily LMWH injections, monitoring at the peak and/or trough, and poorly defined therapeutic targets. Equally surprisingly, we observed a lack of standardization of the anti-Xa activity kits.

## 1. Introduction

Venous thromboembolism (VTE), including deep vein thrombosis and pulmonary embolism, is an important complication in patients hospitalized in intensive care units (ICU) and is associated with increased morbidity and mortality in these particular patients. The risk of VTE in ICU patients is two-fold that of a patient admitted to the general medicine ward. In the observational IMPROVE study including 15,156 medical patients, Spyropoulos et al. [1] reported that admission to an ICU or coronary care unit was a factor independently associated with VTE. Several other factors know to increase the risk of VTE were present in ICU patients: older age, prolonged immobilization due to sedation, mechanical ventilation, central venous catheterization [2], and severe inflammation-like observed during sepsis [3]. The recent pandemic due to coronavirus disease 2019 (COVID-19), where the risk of VTE in hypoxic critically ill patients is even greater, reaffirms the importance of adequate thromboprophylaxis.

In this paper, we review the data from the literature on thromboprophylaxis in ICU patients—first, from a clinical point of view: comparing the administration of low molecular weight heparin (LMWH) to unfractionated heparin (UFH) and, secondly, from a laboratory point of view: the monitoring of these anticoagulants. Studies on COVID-19 patients will not be considered, given the many studies still underway on the subject and the pathophysiology probably being different to explain the high rate of VTE (among others, higher levels of inflammation, endotheliitis, and increased production of neutrophil extracellular traps) [4].

## 2. Thromboprophylaxis in Critically Ill Patients: Clinical Data

Since the Medenox study by Samama et al. [5] comparing enoxaparin at doses of 40 mg once a day and 20 mg once a day versus a placebo in acutely ill medical patients, which showed a significant decrease in the level of VTE for the dose of 40 mg of LMWH versus the placebo, LMWH is recommended in ICU patients, even if this study excluded patients mechanically ventilated or immobilized for three days. Since this study, the dose of 40 mg enoxaparin once a day has been used in the majority of ICU patients, and few new studies on critically ill patients have been performed.

Despite mnemonics by experts of the Critical Care Society (Fast Hug) to encourage teamwork to check some of the key aspects in the general care of all critically ill patients [6] and practice guidelines from scientific societies (for example, the American College of Chest Physicians) that recommends pharmacological VTE prophylaxis in hospitalized patients with high risk of thrombosis, the omission of thromboprophylaxis by clinicians is frequent. Lauzier et al. reported in a retrospective audit performed on 1935 patients in 28 North American ICUs that 15.3% of the patient ICU days (*n* = 12,756) did not receive thromboprophylaxis (neither pharmacological nor mechanical) [7]. The reasons for omission were probably adequate, like a high risk of bleeding (44.5%), current bleeding (16.3%), recent or upcoming invasive procedures (10.2%), and life support limitations (6.9%), but also inadequate: no reason (12.9%) and nighttime admission or discharge (9.7%). Thromboprophylaxis was less often administered to more severe patients: sicker patients, surgical patients, and those receiving vasoactive drugs or renal replacement therapy.

In 2011, the PROTECT study (Prophylaxis for Thromboembolism in Critical Care Trial) compared the effectiveness of the two most common pharmacoprevention strategies against VTE in 3764 medico-surgical critically ill patients: randomization between the administration of LMWH (subcutaneous dalteparin at a dose of 5000 IU once daily plus placebo once daily) or UFH at a dose of 5000 IU twice daily. This study showed no difference in the occurrence of deep venous thrombosis (5.1 versus 5.8% for UFH) but a decrease of the rates of pulmonary embolism (1.3% for LMWH versus 2.3% for UFH; *p* = 0.01) and heparin-induced thrombocytopenia (HIT) in the patients who received LMWH [8]. Moreover, the cost-effectiveness analysis of this study also reported an advantage for LMWH due to the lower rates of pulmonary embolus and HIT [9]. Since the PROTECT and their sub-studies, LMWH remains the cornerstone of the prevention of VTE in critically ill patients, except perhaps in patients with renal dysfunction [10].

Despite “adequate” thromboprophylaxis, several studies performed on critically ill patients reported high incidences of VTE. Beitland et al. reported VTE in 27% of the 70 critically ill patients with a stay longer than 48 h treated with dalteparin 5000 IU/day [11]. Zhang et al. observed a cumulative incidence of VTE at 7, 14, 21, and 28 days of, respectively, 4.45%, 7,14%, 7,53%, and 9.55% in 281 patients despite LMWH 4000 UI twice daily [12]. In 153 critically ill trauma patients, Hamada et al. observed in a prospective observational study an incidence of VTE of 18% despite mechanical and chemical prophylaxis by UFH (5000 UI/12 h) or LMWH (enoxaparin 40 mg/24 h) [13]. In a retrospective study including 355 septic ICU patients for 2 years, Hanify et al. observed that 42 patients (12.5%) developed a VTE despite thromboprophylaxis by UFH or LWMH [14]. In this study, the risk factors of acquiring a VTE were ARDS and higher PEEP and not inflammation, as reported in other studies [4,15]. VTE was associated with an increased ICU and hospital length of stay [14]. In a multicentric international study including 3746 medico-surgical critically ill patients, Lim et al. identified factors that may contribute to failure of the anticoagulant thromboprophylaxis in the occurrence of VTE [16]. Failure was more likely reported in ICU patients with elevated body mass indexes (hazard ratio, 1.18 per 10-point increase; 95% CI, 1.04–1.35; *p* = 0.01), those with a personal or family history of venous thromboembolism (hazard ratio, 1.64; 95% CI, 1.03–2.59; *p* = 0.04), and those receiving vasopressors (hazard ratio, 1.84; 95% CI, 1.01–3.35; *p* = 0.046). These authors concluded that the knowledge of the factors predisposed toward thromboprophylaxis failure and to VTE during critical illness may help clinicians to risk stratify patients and guide the appropriate thromboprophylaxis [16]. Despite “adequate” thromboprophylaxis, VTE occurs in 4–15% of ICU patients, depending also on the routine use of ultrasonography surveillance or not [17]. The exact question all these studies highlighted was to optimize the risk–benefit balance of thromboprophylaxis in critically ill patients.

## 3. LMWH: Schema of Thromboprophylaxis

No real data about the administration schema of LMWH exists for critically ill patients. All of the proposed schemas are extrapolated from other clinical situations. Far before the COVID-19 pandemic situation, a published series showed a shocking amount of disagreement regarding the 40-mg fixed dose of enoxaparin efficacy among various pathologies. This last posology was determined in acute medical hospitalized patients (Medenox) but not for ICU patients [5]. Higher doses and, above-all, weight-adapted doses were then proposed.

The term “intermediate dose” is often used to qualify these regimens situated between a conventional prophylactic and therapeutic regimen. It is often based on a halved therapeutic dose or injection frequency.

Once-daily injections of 85–100 UI/kg/day were initially used in the secondary prevention of venous thromboembolic disease [18] and are now used for hospital and ambulatory primary prevention of high thrombotic-risk patients (pregnant women or cancer patients) [19,20]. In our institution, we recently adopted this schema for patients with COVID-19 inflammatory storms [21].

Twice-daily injections of 50 UI/kg (or 0.5 mg/kg) were first developed in primary prevention in obese patients after bariatric surgery [22,23]. This indication was developed after the observation of the inadequacy of the conventional dosages for this surgery and after the recommendation to adapt the dosages to the weights of these patients, often morbidly obese. Since then, they have been used in other indications, such as in ICU obese trauma patients [24,25]. Some have proposed them as the “standard initial enoxaparin regimen” for ICU patients [26].

### Head-to-Head Comparison of Once- or Twice-Daily LMWH Injections

To date, the only indication where both regimens have been compared is the outpatient treatment of venous thromboembolic disease. Both options seem effective and safe for all LMWH [27]. However, the Riete Registry on enoxaparin pointed out that once-daily injections had more VTE recurrences, fewer major bleeds, and fewer deaths [28]. These protocols left critically ill patients in whom LMWH pharmacokinetics could be altered, and, despite a high anti-Xa peak, for several hours, the anti-Xa activity would be close to zero [29]. Therefore, in situations with a very high thrombotic risk requiring therapeutic anticoagulation, the professional consensus favors the protocols with twice-daily injections. Likewise, for overweight ICU patients with various indications, most modifications in the United States are based on the initial standard 30-mg twice-daily protocol, and the principle of a double injection has not been debated [23,25,30,31].

## 4. Monitoring of Thromboprophylaxis: Peak or Trough or Peak and Trough?

Monitoring LMWHs by anti-Xa activity remains a subject of discussion, and many experts are not in favor of it. As such, the European Society of Anesthesia (European guidelines on perioperative venous thromboembolism prophylaxis) only recommends it as Grade 2C [32].

However, numerous studies have shown that the standard regimens and the very classic “enoxaparin 40 mg/day or 30 mg/12 h” do not allow the levels of anticoagulants considered adequate to reach. For example, Robinson et al. reported a 3-day study on enoxaparin pharmacokinetic in nonobese medico-surgical ICU patients. In this study, four regimens (40-mg QD, 30-mg BID, 40-mg BID, and 1-mg/kg QD) were compared for their ability to maintain the anti-Xa activity between 0.1 and 0.4 IU/mL (samples at the baseline, 4, 12, 16, and 24 h). Eighty patients ranging from 50 to 90 kg in weight were enrolled. The anti-Xa activity was within the target for 33.3% (40-mg QD), 41.7% (30-mg BID), 83% (1-mg/kg QD), and 91.7% (40-mg BID) of the study period [33]. These data confirmed the inadequacy of the standard protocols, especially for more severe or obese patients.

Some studies have been able to establish a link between low levels of LMWH measured by anti-Xa activity and the occurrence of thrombotic events. Most often, LMWHs are assessed at the peak, 4 h after injection, as the peak anti-Xa is considered a more useful predictor of LMWH safety and efficacy than the trough. For instance, in trauma patients, a study comparing enoxaparin 40-QD versus BID showed a marked increase in the incidence of DVT when the peak anti-Xa activity was low (22% versus 7%), regardless of the posology [34].

However, the trough levels also seem to have a clear clinical impact. Previous research involving VTE prophylaxis after hip replacement surgery found that patients with 12-h anti-Xa levels of less than 0.1 IU/mL experienced a significantly higher rate of VTE (15% versus 6%, *p* = 0.05), and levels exceeding 0.2 IU/mL were associated with higher rates of wound hematoma formation (24.5% versus 5.3%, *p* <0.01) [35]. Other studies have correlated low trough levels with a higher incidence of VTE [36,37]. This was confirmed by Malinoski et al. in a study on 54 patients where <0.1 IU/mL trough levels, but not the peak levels, were associated with more VTE than those with anti-Xa trough levels >0.1 IU/mL (37 versus 11%, *p* = 0.026) [38]. There is not always a relationship between the anti-Xa peak and trough, and perhaps both need to be evaluated to optimize antithrombotic prophylaxis [39,40].

In fact, peak and trough monitoring each have specific problems. For the peak, it seems that a correct sampling time is difficult to respect [41]. Moreover, some factors can induce a variation in the moment when the activity peak is reached, as one versus two daily doses (3 to 4 h versus 4–6 h) or can alter the subcutaneous reabsorption.

The trough levels are frequently near 0 IU/mL, and these low levels are difficult to assess: according to the External Quality Control data, the precision decreases significantly below 0.35 IU/mL [42], and some laboratories indicate a lower limit of quantification at 0.1 or even 0.2 IU/mL [23]. The test setups for heparin monitoring were initially developed for peak measurements. It is only with the development of Direct Oral Anticoagulants that a real interest in trough measurements appeared, being the best indicator for accumulation (stated, e.g., by Ludwig et al. [23]).

However, it seems possible, without major modifications, to obtain very satisfactory trough results. In our laboratory, we observed over a 2-month period with STA-Liquid Anti-Xa (Stago, Gennevilliers, France) a standard deviation of 0.01 IU/mL for a pooled normal plasma (null anti-Xa activity). The problem seems rather to come from the calibration curves—often very broad, with a measurement range from zero to ~2 IU/mL. A minimal loss of linearity (usually linear HEMI log) may induce a significative bias at the zero point (up to 0.05 IU/mL, in our experience). However, this can be compensated by a specific low curve (e.g., limited to 1 UI/mL) or by using a polynomial instead of linear best fit.

## 5. UFH in ICU

Although unfractionated heparin (UFH) has been replaced by LMWH for many reasons, it remains the anticoagulant of choice among selected patients: during extracorporeal membrane oxygenation (ECMO), patients with severe renal failure (clearance of creatinine <30 mL/min^2^), or patients in whom a rapid reversal of the anticoagulant effect may be required (atrial fibrillation, VTE, invasive procedure, etc.).

UFH may be used at high, intermediate, or low (prophylactic) doses; however, evidence-based data remains limited. For prophylaxis, the most common dosage is 5000 U/12 h, but this seems frequently insufficient, especially in obese patients, and dosages of 5000 U/8 h (King et al. for a review on medical patients) [43] or 8000 U/12 h have been proposed after bariatric surgery [44].

An adjustment of these doses is sometimes performed with APTT or anti-Xa activity. In a study on bariatric surgery, a target of 0.11–0.25 IU/mL was proposed [44]. In another study, the same team proposed a continuous infusion protocol, which had the advantage of eliminating many factors of variability, with a target of 0.15–0.25 IU/mL [45]. This must be compared with the ACCP guidelines for intermediate anticoagulation in pregnant women, which propose a target of 0.1–0.3 IU/mL [46]. During an internal consensus meeting in our institution, a target of 0.3–0.5 was proposed as an intermediate UFH target for thromboprophylaxis in COVID-19 patients (unpublished).

### 5.1. UFH Monitoring and APTT: The Too-Much Job?

The Activated Partial Thromboplastin Time (APTT) is not a reflection of UFH activity but a surrogate marker of its concentration. The APTT response to UFH varies with the reagent composition (activator, phospholipid nature, and concentration) and clot system detection (mechanical, optical with various wavelengths, and detection algorithm). To minimize some of these variations, the UFH therapeutic range (HTR) is usually communicated as the ratio of the normal pooled plasma. The only valid way to determine this for a specific analytical system (one lot of reagents and one type of analyzer) is to establish the best fit line comparing the APTT result and anti-Xa activity of the ex vivo samples of patients under UFH therapy and to calculate the APTT ratios for the upper and lower therapeutic range (0.3–0.7 IU/mL) [47]. Any other intermediate or prophylactic zone should be established according to the same procedure. Thereafter, the reagent is classified as lowly (HTR = 1.5–2.5 ratio) or highly (HTR = 2.0–3.0 or 3.5) sensitive to heparin. The latter are usually preferred for UFH monitoring [48]. Therefore, “crude APTT values” in seconds are not adequate to communicate an APTT HTR. These procedures, established more than 20 years ago, are unfortunately little-respected.

However, even after lab-specific HTR determination, the correlation between the APTT and UFH anti-Xa activity remains poor. In particular, an increase of acute-phase proteins like factor VIII may falsely reduce the UFH APTT sensitivity, leading to supra therapeutic UFH levels. Therefore, APTT is not adequate for UFH monitoring in patients with a major inflammatory syndrome like COVID-19 [49], despite the other problems with anti-Xa activity. However, with a heparin high-sensitivity APTT, it seems possible to do a good job. Figure 1 illustrates the relationship between anti-Xa and APTT in our institution in COVID-19 patients. No heparin resistance seemed to be present, as the D-dimers, fibrinogen, CRP, and platelets level had no influence on this relationship.

### 5.2. UFH Monitoring and Anti-Xa Activity

The monitoring of anticoagulants with anti-Xa activity has become widely available with a new generation of ready-to-use kits at very acceptable costs. Excess Factor Xa is added to the test plasma and is inhibited by an anticoagulant. Residual Factor Xa cleaves a synthetic chromogenic substrate. All anticoagulants with anti-Xa activity, either after binding to antithrombin (UFH, LMWH, Danaparoid, and Fondaparinux) or directly, (anti-Xa DOACS) can be assayed with these kits, but each time needs a specific setting and the appropriate calibration.

Being able to test a wide variety of anticoagulants from a single kit is a clear advantage for laboratory management but also a potential source of error. If there is an error in the identity of the anticoagulant to be tested (e.g., one DOAC for another), the results may mislead the clinician. To reduce this risk, in our laboratory, any request for anticoagulant testing must be done with the international nonproprietary name. Likewise, in the event of a change of anticoagulant, the results of the dosage of the new anticoagulant will only be valid after elimination of the old one. In a patient with renal insufficiency, this may take far more than 24 h. This type of problem is, for example, encountered in patients treated with LMWH and then shifted to unfractionated heparin for renal failure.

For the switch from anti-Xa DOAC to heparin, heparin can be assessed after DOAC elimination by filtration or adsorption procedures [50,51]. The residual concentration of the DOAC can be assessed by some anti-Xa activity kits that have been adapted with specific buffers that block the influence of heparin and its derivatives, thus allowing DOAC determination [52]. 

### 5.3. Heparin Resistance and AT and Dextran Discussions

Heparin resistance can be defined as the need for more than 35,000 IU/day to prolong activated PTT in the therapeutic range or the impossibility of doing so. Several mechanisms may lead to real or apparent heparin resistance [53]. Real heparin resistance may be due to a decreased AT level; PF4 liberation (due to platelet activation or PF4 liberation from endothelium); or aspecific heparin binding to various proteins (vitronectin, fibronectin, Protein S, kininogen, etc.).

Facing this problem, the assay of heparin anti-Xa activity is considered the gold standard. However, there are differences between these assays regarding the presence or absence of exogenous antithrombin (AT), as well as the presence or absence of additional dextran sulphate (DS), and these differences in composition will induce result discrepancies in these situations.

To date, kits without exogenous AT seem to be preferred, as they more accurately reflect the true heparin activity. However, clinical discrepancies seem only to appear for a very marked AT deficiency (<40%) [54].

Aspecific heparin binding may induce discordance between APTT and some anti-Xa assays. Indeed, some reagents contain dextran sulfate (DS), whereas other do not. This is added mainly to prevent ex vivo heparin neutralization by platelet factor 4 (PF4) released from the platelets. The presence of DS will liberate PF4-bound heparin, and these assays give higher anti-Xa activity than those that do not contain it [55]. In low values (~0.2 IU/mL), the difference can rise from simple to double [42]. Likewise, the APTT reagent will not be sensitive to PF4-bound heparin.

Heparin–protein interactions also occur in vivo and are increased in patients with a major inflammatory syndrome, like COVID-19 patients. [56]. The role of this reversible pool of non-AT-linked heparin and the effect of DS on it remain little explored and a subject of discussion. A close phenomenon has been described after the injection of protamine sulphate, where a reagent containing DS gave higher results than expected compared to classical coagulometric tests [57].

True “false” or “apparent” heparin resistance is mainly due to a loss of sensitivity of the APTT reagent to the presence of heparin. This seems mainly due to an elevated concentration of F VIII, a frequently observed phenomenon, and numerous authors have considered that APTT is inadequate to monitor UFH in these conditions [58], but this point is still disputed [59]. To date, the lack of standardization between the anti-Xa UFH assays is still an active and unresolved problem [60].

This problem seems less critical for LMWHs, which have much weaker aspecific interactions, but since most of the tests are performed with enoxaparin, a very low molecular weight heparin (4500 Daltons), this should be confirmed with a higher molecular weight LMWH such as tinzaparin (6500 Daltons).

## 6. Conclusions

COVID-19 let us discover a pathology clearly associated with overweightness, and the inadequacy of a fixed dose of anticoagulants was suddenly imposed, even though we have been talking about it for more than 10 years. For both UFH and LMWH, three therapeutic areas seem to be defined (prophylactic, intermediate, and therapeutic), but the simple question of one or two injections per day has still not been evaluated in the most instances.

However, for antithrombotic prophylaxis in ICU patients with a major inflammatory syndrome, there is strong evidence that an intermediate dose of LMWH should be considered first. Several arguments, not refuted, plead for a regimen of two daily injections, and this one is currently the best-known, with validated therapeutic targets. As have others, we therefore suggest starting with twice-daily 40–50-IU/kg injections and, if the stay is longer than 4 days, to check the anti-Xa therapeutic targets, including the trough levels.

There are too many variables that influence the APTT for them to serve as a basis for multicentric evaluations. However, the test that serves as the basis for this discussion, anti-Xa activity, remains poorly standardized, with clinically significant differences according to the manufacturer. It is therefore necessary to quickly resolve this problem, considering, primarily, the request of clinicians: reliable active heparin monitoring.

## Figures and Tables

**Figure 1 biomedicines-09-00864-f001:**
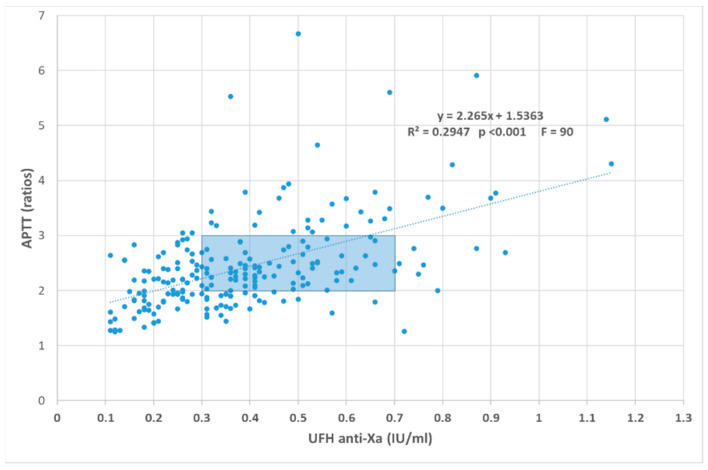
Real-world relationship between UFH anti-Xa (Liquid anti-Xa, Stago) and STA-PTT-A (Stago), a highly sensitive to heparin APTT reagent, in 218 samples from ICU COVID-19 patients. Therapeutic consensus area is highlighted. The D-dimers, fibrinogen, CRP, and platelets levels have no influence on this relationship (comparison of the data pairs for which the parameter of interest (D-dimers, fibrinogen, CRP, and platelets) is, respectively, below the 25th or above the 75th percentiles).

## Data Availability

Not applicable.

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
