# Peer review of "What Do We Know about Thromboprophylaxis and Its Monitoring in Critically Ill Patients?"

_biomedicines, 2021, doi:10.3390/biomedicines9080864_

Round 1

Reviewer 1 Report

Cauchie and Piagnerelli propose an interesting review of what is known about thromboprophylaxis in ICU patients and what remains controversial. This particular topic comes back into the light with the COVID-19 pandemic. They develop and discuss the occurrence and recurrence of VTE in ICU patients, the LMWH and UFH doses and schemes of administration, the laboratory monitoring strategies and the areas that remain to be optimized. Some concerns warrant further explanation and attention by the authors.

1/- In the introduction, the authors announce that studies in COVID-19 patients were not considered given the significant number of ongoing studies and the potential different pathophysiology of VTE in this context compared with non-COVID-19 ICU patients. This is despite the fact that this particular topic comes back into the light because of the COVID-19 pandemic. However, the unique figure of the manuscript reports original results of the relation between UFH anti-Xa activity and aPTT values in 218 samples from COVID-19 ICU patients. This is somewhat weird. Adding some clinical data on VTE and thromboprophylaxis in COVID-19 ICU patients would be a tremendous asset for the manuscript.

2/- The authors discussed the relation between UFH anti-Xa activity and aPTT. What about the relation between LMWH anti-Xa activity and aPTT?

3/- Similarly, the authors tackled the issue of LMWH monitoring at “peak or trough” or “peak and trough”. What about UFH monitoring when given subcutaneously? Should it be monitored at peak or trough?

4/- D-dimers, fibrinogen, CRP and platelets level have no influence on the relation between UFH anti-Xa activity and aPTT values. How this influence was evaluated? It will be wise to add a table reporting the corresponding statistical analysis results.

5/- Please add the “p” value to the Figure 1 in addition to the reported R2.

6/- In page 6, the authors gave the example of the interference of LMWH with UFH anti-Xa activity measurement when patients are switched from LMWH to UFH therapy. Then they mentioned the devices developed to remove DOAC from plasma samples. It is somewhat confusing. This part needs to be modified and made clearer.

7/- The authors mentioned the possibilities to remove DOAC by filtration or absorption. Ref 50 is for DOAC adsorption. Therefore, I suggest to change “absorption” to “adsorption” and add a reference to DOAC filtration, such as “C. Farkh et al. A Diagnostic Solution for Lupus Anticoagulant Testing in Patients Taking Direct Oral FXa Inhibitors Using DOAC Filter. Front Med 2021. DOI: 10.3389/fmed.2021.683357”

8/- Minor points:

  • Page 2, L90-91: “three thousand seven hundred forty-six” to be changed to “3746”
  • Page 3, L104: “really data” to be changed to “real data” or “really reliable data”
  • Page 3, L105: “all of them purposed are …” to be rephrased
  • Page 3, L107: “various of pathologies” to be changed to “various pathologies”
  • Page 3, L143: “have been able to show” to be changed to “have shown”
  • Page 4, L170-171: “For the peak, many … is reached” to be rephrased
  • Page 4, L194: “mL/min2” to be corrected
  • Page 5, L206: “proposes” to be changed to “propose”
  • Page 5, L221: “latter” to be changed to “latters”
  • Page 6, L288: “Tinzaparine” to be changed to “tinzaparin”
  • Page 7, L297: “the test which as to serve as a basis for discussion” to be rephrased
  • Page 7, L298-299: “with very significant differences according to the manufacturer” to be rephrased
  • Page 7, L300: “first” to be changed to “primarily”

Author Response

Dear reviewer 1

We thank you for the relevant comments and most of your suggestions have been followed. However, for 3 of them we allow ourselves to come back to you.

1/ The only purpose of the figure 1 is to illustrate the relationship between APTT and anti-Xa activities in patients with major inflammatory syndrome, as COVID-19 patients. These patients have been indicative of an inadequate thromboprophylaxis pattern, but they are not the topic of our manuscript. Starting a discussion on the particular pathophysiology of COVID-19 (endotheliitis, increased production of extracellular neutrophil traps,…) would be long and confusing.

2/This relationship between LMWH anti-Xa activity and aPTT varies over time following the more rapid loss of long fragments (with anti-IIa activity) and does not seem to be of interest.

3/The use of subcutaneous UFH is very problematic in patients with major inflammatory syndrome and large fluctuations in subcutaneous resorption and pharmacokinetic. Many think that these fluctuations mean that in practice, this type of treatment is not really monitorable and thus not recommended in this particular population

Reviewer 2 Report

Content suggestions:

  1. I would like to kindly ask the authors to include the discussion about the role of monitoring of D-dimers in the patients hospitalized in ICU receiving LMWH and UFH.
  2. The manuscript does not contain any resulting recommendation for practice. Thus, from the practical point of view, I would like to kindly ask the authors to prepare some suggestions.

Author Response

Dear reviewer 2

We thank you for the relevant comments and we add a paragraph in the conclusion.

Concerning D-dimers concentrations, they are little used in our ICUs because they have no direct impact on patient care. They are only mentioned in our manuscript in the APTT vs UFH relationship as a marker of coagulation activation. For COVID-19 patients, they were mainly used as a prognostic parameter on admission, for the calculation of the DIC score or to diagnose venous thromboembolism diseases.

Round 2

Reviewer 1 Report

The authors responded appropriately to all the points I had raised. I have no more additional concern.

Author Response

.